# OBJECT REPRESENTATIONS AS FIXED POINTS: TRAINING ITERATIVE INFERENCE ALGORITHMS WITH IMPLICIT DIFFERENTIATION

**Michael Chang, Sergey Levine & Thomas L. Griffiths** *

## ABSTRACT

Deep generative models, particularly those that aim to factorize the observations into discrete entities (such as objects), must often use iterative inference procedures that break symmetries among equally plausible explanations for the data. Such inference procedures include variants of the expectation-maximization algorithm and structurally resemble clustering algorithms in a latent space. However, combining such methods with deep neural networks necessitates differentiating through the inference process, which can make optimization exceptionally challenging. We observe that such iterative amortized inference methods can be made differentiable by means of the implicit function theorem, and develop an implicit differentiation approach that improves the stability and tractability of training such models by decoupling the forward and backward passes. This connection enables us to apply recent advances in optimizing implicit layers to not only improve the stability and optimization of the slot attention module in SLATE, a state-of-the-art method for learning entity representations, but do so with constant space and time complexity in backpropagation and only one additional line of code.

## 1 INTRODUCTION

A major goal of building human-level artificial intelligence is that of replicating how humans abstract experience into coherent entities that are used for high-level reasoning. A concrete setting for studying this question has been in so-called **object-centric learning**, which seeks to decompose observations $x$ into a set of independent representations of entities without supervision on how to decompose. Each datapoint $x^n$ (e.g. image) is modeled as a set of independent sensor measurements $x^{n,m}$ (e.g. pixels) which are generally posited as having been generated from a mixture model whose components represent the entities. Under a clustering lens, the problem reduces to finding the $K$ groups of cluster parameters $\theta^n := \{\theta^{n,k}\}_{k=1}^K$ and cluster assignments $\phi^{n,m} := \{\phi^{n,m,k}\}_{k=1}^K$ that were responsible for the measurements $x^{n,m}$ of the datapoint $x^n$. Modeling entities as cluster components encodes the assumption that entities are *independent* and *symmetric*, thereby requiring a mechanism for breaking symmetry equally valid explanations of the same observation.

Thus inference is typically done by breaking symmetry via a set of random initial guesses

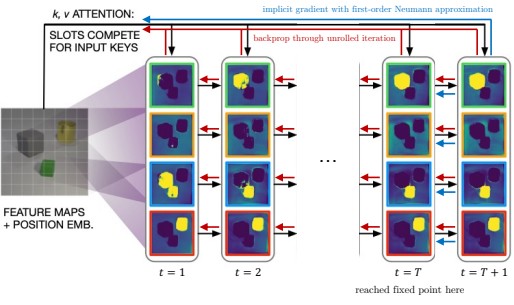

(a) Implicit differentiation of slot attention

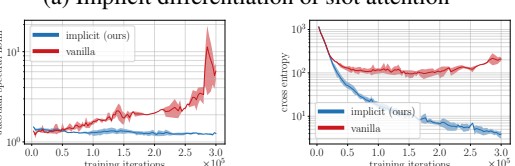

(b) Jacobian norm      (c) Validation loss

Figure 1: **Overview.** We propose to train the slot attention model (1a), whose figure is adapted from Locatello et al. (2020), with implicit differentiation. Our approach leads to more stable training (1b) and substantially lower validation loss (1c) compared to vanilla slot attention.

---

*mbchang@berkeley.edu, svlevine@eecs.berkeley.edu, tomg@princeton.edu

$\theta^{n,k}$ and then iteratively updating $\boldsymbol{\theta}^n$ and $\phi^{n,m}$

during the course of execution. Methods that learn the update rule as a deep network $f_{\mathbf{w}}$ are instances of **iterative amortized inference** (Marino et al., 2018). The state-of-the-art slot attention module (Locatello et al., 2020), e.g., computes $\boldsymbol{\theta}_{t+1}^n \leftarrow f(\boldsymbol{\theta}_t^n, x^n)$, where $\phi^{n,m}$ is updated as an intermediate step inside $f_{\mathbf{w}}$. The $\boldsymbol{\theta^n}$, called *slots*, serve as input to a downstream objective, e.g. image reconstruction. Other instances of this approach include (Greff et al., 2017; Van Steenkiste et al., 2018; Greff et al., 2019; Veerapaneni et al., 2020; Locatello et al., 2020; Kipf et al., 2021; Zoran et al., 2021; Singh et al., 2021). All differentiate through the unrolling of $f_{\mathbf{w}}$.

Despite their conceptual elegance, it has been difficult to scale such iterative amortized inference methods beyond modeling simple static scenes or short video sequences because differentiating through the unrolled forward iteration makes training unstable. Fig. 5 shows that the spectral norm of the Jacobian of $f_{\mathbf{w}}$ gradually increases over the course of training, which has empirically been observed to cause training instabilities (Bai et al., 2021). Such instabilities result in sensitivity to hyperparameter choices (e.g., number of inference iterations) and have motivated adding optimization tricks such as gradient clipping, learning rate warm-up, and learning rate decay, all of which make such models more complex and harder to use, restrict the model from optimizing its learning objective fully, and only temporarily delay instabilities that still emerge in later stages of training.

To solve this problem, we observe that previous methods have not taken advantage of the fact that $f_{\mathbf{w}}$ can be viewed as a fixed point operation. Thus, $f_{\mathbf{w}}$ can be trained with **implicit differentiation** applied at the fixed point, without backpropagating gradients through the unrolled iterations. *Our primary contribution is to propose implicit differentiation for training the iterative amortized inference procedures of symmetric generative models, such as those used for learning object representations.* Specifically, we show across three datasets that, compared to the latest state-of-the-art of these methods, SLATE (Singh et al., 2021), our method for training SLATE achieves much lower validation loss in training, as well as lower Fréchet inception distance (FID) (Heusel et al., 2017) and mean squared error (MSE) in image reconstruction. Our method also removes the need for gradient clipping, learning rate decay, learning rate warmup, or tuning the number of iterations, while achieving lower space and time complexity in the backward pass, all with just one additional line of code.

## 2 BACKGROUND

Implicit differentiation is a technique for computing the gradients of a function defined in terms of satisfying a joint condition of the input and output. For example, a fixed point operation $f$ is defined to satisfy "find $\lambda$ such that $\lambda = f(x, \lambda)$," rather than through an explicit parameterization of $f$. This fixed point $\lambda_*$ can be computed by simply repeatedly applying $f$ or by using a black-box root-finding solver. Letting $f_{\mathbf{w}}$ be parameterized by weights $\mathbf{w}$, with input $x$ and fixed point $\lambda_*$, the implicit function theorem (Cauchy, 1831) enables us to directly compute the gradient of the loss $\ell$ with respect to $\mathbf{w}$, using only the output $\lambda_*$:

$$\frac{\partial \ell}{\partial \mathbf{w}} = \frac{\partial \ell}{\partial \lambda_*} \left(I - J_{f_{\mathbf{w}}}(\lambda_*)\right)^{-1} \frac{\partial f_{\mathbf{w}}(\lambda_*, x)}{\partial \mathbf{w}}, \quad (1)$$

where $J_{f_{\mathbf{w}}}(\lambda_*)$ is the Jacobian matrix of $f_{\mathbf{w}}$ evaluated at $\lambda_*$. Compared to backpropagating through the unrolled iteration of $f$, which is just one of many choices of the solver, implicit differentiation via Eq. 1 removes the memory cost of storing any intermediate results from the unrolled iteration.

Much effort has been put into approximating the inverse-Jacobian term $\left(I - J_{f_{\mathbf{w}}}(\lambda_*)\right)^{-1}$ which has $\mathcal{O}(n^3)$ complexity to compute. Geng et al. (2021); Fung et al. (2021); Huang et al. (2021); Shaban et al. (2019) propose instead to approximate $\left(I - J_{f_{\mathbf{w}}}(\lambda_*)\right)^{-1}$ with its Neumann series expansion:

$$\left(I - J_{f_{\mathbf{w}}}(\lambda_*)\right)^{-1} = \lim_{T \to \infty} \sum_{i=0}^{T} J_{f_{\mathbf{w}}}(\lambda_*)^i. \quad (2)$$

The first-order approximation ($T = 1$) amounts to applying $f$ once to the fixed point $\lambda_*$ and differentiating through the resulting computation graph. This is not only cheap to compute and easy to implement, but has also been shown empirically (Geng et al., 2021) to have a regularizing effect on the spectral norm of $J_{f_{\mathbf{w}}}$ without sacrificing performance.

## 3  IMPLICIT OBJECT-CENTRIC LEARNING

By recognizing that slot attention implements a fixed point operation, we propose **implicit slot attention**: a method for training the state-of-the-art slot attention module (Locatello et al., 2020), with the simplest and most effective method that we have empirically found for approximating the implicit gradient, which is its first-order Neumann approximation, although in principle any black box solver for computing the fixed point and black box gradient estimator for computing the implicit gradient can be used. It can be implemented by simply differentiating the computation graph of applying the slot attention update *once* to the fixed point $\theta_*^n$, where $\theta_*^n$ is computed by simply iterating the slot attention module forward as usual, but *without* the gradient tape. The time and space complexity of backpropagation for our method compared to vanilla slot attention as a function of the number of slot attention iterations $n$, is shown below:

```
def step(slots, k, v):
    # compute assignments given slots
    q = project_q(norm_slots(slots))
    k = k * (slot_size ** (-0.5))
    attn = F.softmax(torch.einsum('bkd,bqd->bkq', k, q), dim=-1)
    attn = attn / torch.sum(attn + epsilon, dim=-2, keepdim=True)
    # update slots given assignments
    updates = torch.einsum('bvq,bvd->bqd', attn, v)
    slots = gru(updates, slots)
    slots = slots + mlp(norm_mlp(slots))
    return slots

def iterate(f, x, num_iters):
    for _ in range(num_iters):
        x = f(x)
    return x

def forward(inputs, slots):
    inputs = norm_inputs(inputs)
    k, v = project_k(inputs), project_v(inputs)
    slots = iterate(lambda z: step(z, k, v), slots, num_iterations)
    slots = step(slots.detach(), k, v)
    return slots
```

Figure 2: **Code.** The first order Neumann approximation to the implicit gradient adds only one additional line of Pytorch code (Paszke et al., 2019) to the original forward function of slot attention, but yields substantial improvement of optimization. `attn` and `slots` correspond to $\phi$ and $\theta$ in the text respectively.

|  | vanilla slot attention | ours |
|---|---|---|
| time (forward) | $\mathcal{O}(n)$ | $\mathcal{O}(n)$ |
| space (forward) | $\mathcal{O}(n)$ | $\mathcal{O}(n)$ |
| time (backward) | $\mathcal{O}(n)$ | $\mathcal{O}(1)$ |
| space (backward) | $\mathcal{O}(n)$ | $\mathcal{O}(1)$. |

Our method is not only more efficient but also requires only one additional line of code (Fig. 2).

## 4  EXPERIMENTS

The main hypothesis behind this paper is that implicit differentiation can improve the training of iterative amortized inference methods for object-centric learning. We test this hypothesis by replacing the backward pass of the slot attention module in SLATE (Singh et al., 2021) with the first-order Neumann approximation of the implicit gradient, and measuring optimization performance.

For the task of image reconstruction, SLATE uses a discrete VAE (Ramesh et al., 2021) to compress an input image into a grid of discrete tokens. These tokens index into a codebook of latent code-vectors, which, after applying a learned position encoding, serve as the input to the slot attention module. An Image GPT decoder (Chen et al., 2020) is trained with a cross-entropy loss to autoregressively reconstruct the latent code-vectors, using the outputted slots from slot attention as queries and the latent code-vectors as keys/values. Gradients are blocked from flowing in and out of the discrete VAE to the rest of the network (i.e. the slot attention module and the Image GPT decoder), but the entire system is trained simultaneously.

We consider three datasets: CLEVR-Mirror (Singh et al., 2021), Shapestacks (Groth et al., 2018), and COCO-2017 (Lin et al., 2014). We obtained CLEVR-Mirror directly from the SLATE authors and used a 70-15-15 split for training, validation, and testing. We pooled all the data variants of Shapestacks together as Singh et al. (2021) did and used the original train-validation-test splits. The COCO-2017 dataset was downloaded from FiftyOne and used the original train-validation-test splits.

### 4.1  DOES IMPLICIT DIFFERENTIATION STABILIZE THE TRAINING OF SLOT ATTENTION?

Using the two primary metrics used in Singh et al. (2021), images generated by SLATE trained with implicit differentiation achieve both lower pixel-wise mean-squared error and FID score (Heusel et al., 2017). The FID score was computed with the PyTorch-Ignite (Fomin et al., 2020) library using the inception network from the PyTorch port of the FID official implementation. All methods were trained

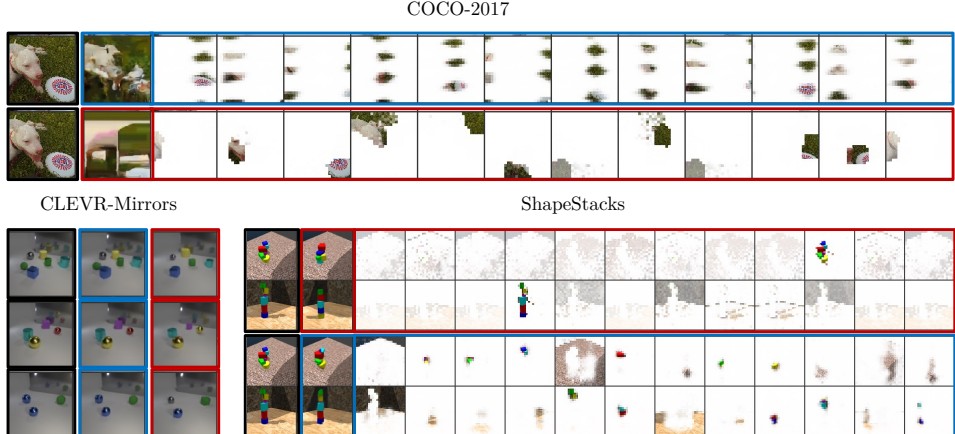

Figure 3: **Qualitative results.** Across three datasets, optimizing SLATE with implicit differentiation leads to improved image reconstructions through the slot bottleneck. Black borders indicate the ground truth image. Blue border indicate our method. Red borders indicate vanilla SLATE. The rest of the panels show attention masks.

for 250k gradient steps. Table 1 compares the FID and MSE scores of the images that result from compressing the SLATE encoder's set of discrete tokens through the slot attention bottleneck, using Image-GPT to autoregressively re-generate these image tokens one by one, and using the discrete VAE decoder to render the generated image tokens. Implicit differentiation significantly improves the quantitative image reconstruction metrics of SLATE across the test sets of CLEVR-Mirrors, Shapestacks, and COCO. In the case of MSE for CLEVR, this is almost a 7x improvement.

The higher quantitiatve metrics also translate into better quality reconstructions on the test set, as shown in Figure 3. For CLEVR-Mirrors, vanilla SLATE sometimes drops or changes the appearance of objects, even simple scenes with three objects. In contrast, the reconstructions produced from training with implicit differentiation match the ground truth very closely. For Shapestacks, our method consistently segments the scene into constituent objects. This is sometimes the case with vanilla SLATE on the training and validation set as well, but we observed

Table 1: Quantitative metrics for image reconstruction through the slot bottleneck.

| Data | Ours | Vanilla |
|------|------|---------|
| CLEVR (FID) | **22.19** | 25.89 |
| CLEVR (MSE) | **10.66** | 67.04 |
| COCO (FID) | **127.79** | 147.48 |
| COCO (MSE) | **1659.15** | 1821.75 |
| ShapeStacks (FID) | **34.2** | 34.76 |
| ShapeStacks (MSE) | **108.67** | 312.14 |

for both of the seeds we ran that vanilla SLATE produced degenerated attention maps where one slot captures the entire foreground, and the background is divided among the other slots. The visual complexity of the COCO dataset is much higher than either CLEVR-Mirrors and Shapestacks, and the reconstructions on the COCO dataset are quite poor, for both SLATE's discrete VAE and consequently for the reconstruction through the slot bottleneck. This may be expected because we did not attempt to tune SLATE's hyperparameters to COCO, but it does highlight the gap that still exists between using the state-of-the-art in object-centric learning out-of-the-box and what the community may want these methods to do. The attention masks for both the vanilla SLATE and our method furthermore do not appear to correspond consistently to coherent objects in COCO but rather patches on the image that do not immediately seem to match with our human intuition of what constitutes a visual entity.

## 4.2 CAN WE SIMPLIFY THE NEED FOR OPTIMIZATION TRICKS?

To further understand the benefits of implicit differentiation, we then ask whether it stabilizes the training of slot attention without the need for optimization tricks like learning rate decay, gradient clipping, and learning warmup. Fig. 4 shows that these tricks generally help regularize spectral norm of the Jacobian of vanilla slot attention but are not required by our method. Decaying the learning rate regularizes the Jacobian norm from exploding, but it also hurts optimization performance for both our method and vanilla SLATE, as expected. When we remove gradient clipping, the Jacobian norm

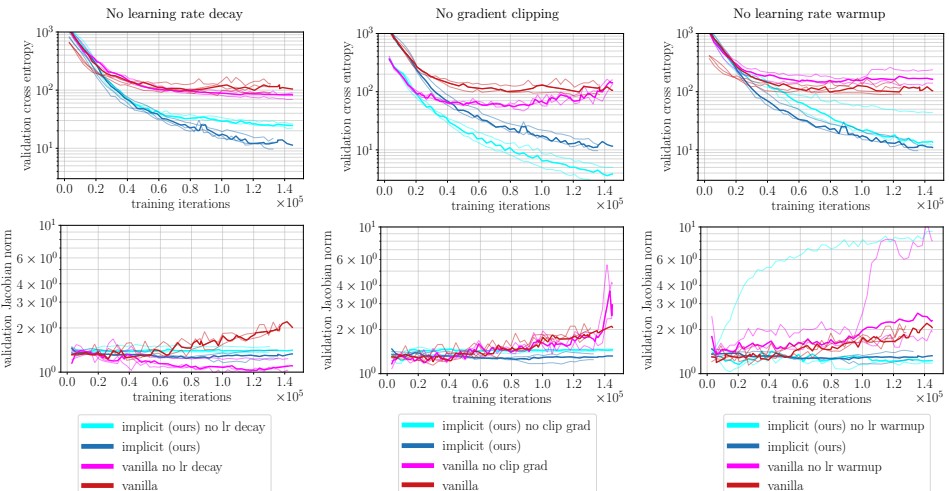

Figure 4: **Implicit differentiation removes the need for many optimization tricks.** We ablate three heuristic optimization tricks from both vanilla SLATE and our method. Whereas removing gradient clipping and learning rate warmup causes vanilla SLATE's training to become unstable, as indicated by the growth of the Jacobian norm, our method trains more stably and can take advantage of larger gradient steps.

of vanilla SLATE explodes, as do its gradients (Fig. 5a), whereas both stay stable for our method. Lastly, removing learning rate warmup also consistently makes vanilla SLATE's training unstable, whereas it only affects the stability of our method for one out of three seeds. Finally, Fig. 5b shows that our method is not sensitive to the number of iterations with which to iterate the slot attention cell, whereas vanilla slot attention is, with more iterations being harder to train.

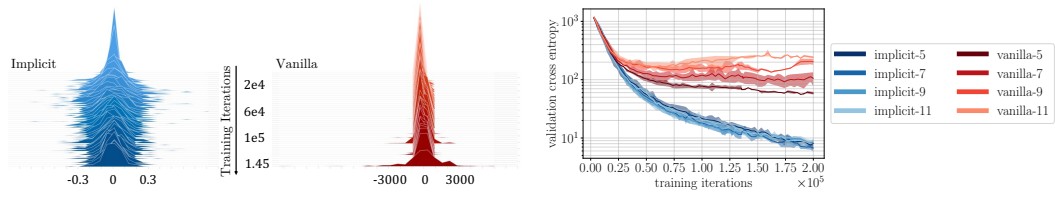

(a) No need for gradient clipping.

(b) No sensitivity to number of iterations.

Figure 5: **Stability.** (5a) Without gradient clipping, our implicit differentiation technique keeps gradients small while backpropagating through the unrolled iterations causes gradients to explode. (5b) Training with implicit differentiation also is not sensitive to the number of iterations with which to iterate the slot attention module.

## 5 DISCUSSION AND LIMITATIONS

We have proposed implicit differentiation for training the iterative amortized inference procedures of symmetric generative models and demonstrated this technique on a state-of-the-art object-centric learning method. Our results show clear signal that implicit differentiation can offer a significant optimization improvement over backpropagating through the unrolled iteration of slot attention, and potentially any iterative inference algorithm, with lower space and time complexity and only one additional line of code. Despite our work pushing the optimization performance for a state-of-the-art model in object-centric learning, the discrepancy between the quantitative improvement in optimization and evaluation metrics on the one hand and the less intuitive qualitative attention masks on real world observations like COCO (Fig. 3) on the other hand still suggests a gap between what we optimize these methods to do and what we actually want them to do. This paper proposes a novel conceptualization of object representations as fast weights that converge towards a set of fixed points during execution. Because it is so simple to apply implicit differentiation to any fixed point algorithm,

we hope this work inspires future work to leverage tools developed for implicit differentiation for improving object-centric learning and methods for learning latent structure more broadly.

## ACKNOWLEDGEMENTS

This work was supported by ARL, W911NF2110097, with computing support from Google Cloud Platform.

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
