# OpenReview forum: "Object Representations as Fixed Points: Training Iterative Inference Algorithms with Implicit Differentiation"
_ICLR.cc/2022/Workshop/OSC — ICLR2022 OSC  Oral_

### Official Review · Reviewer_UmAK · 2022-03-10
**Stabilizes Slot Attention training via implicit differentiation**

**Rating:** 2
**Confidence:** 3

**Review:**

Authors argue that combining iterative inference procedures (e.g. EM) with neural networks is challenging due to differentition through the inference process. They propose to make amoritzed iterative inference differentiable by implicit function theorem. This implicit differentiation decouples forward and backward pass and improves stability and tractability. They experiment with slot attention module in SLATE.

### General comments:
Typo in paragraph 2 of the introduction: "e.g., n computes θt+1 n ← f (θt n, xn )", should it be f_w?

Table 1 would've been more readable if it had 5 (1+2x2) columns, Data | FID_ours | FID_slate | MSE_ours | MSE_slate .

I think the title 4.1 is wrong. It states "does implicit differentiation stabilize the training of slot attention?", but then talks about performance improvements. On the other hand, section 4.2 talks about training stabilization, namely it's first sentence is: "To further understand the benefits of implicit differentiation, we then ask whether it stabilizes the training of slot attention without the need for optimization tricks like learning rate decay, gradient clipping, and learning warmup."

Figure 4 is unclear what the shaded lines mean.

### Pros:
Shows that implicit differentiation can stabilize the training and reduce the need for optimization "tricks".
The paper definitely fits the workshop theme, is technically correct, novel and clearly written.

### Cons and room for improvement:
I'm a bit skeptical that whether all the optimization tricks could be removed, as the authors only ablated by by removing individual tricks (figure 4). It would also be nice to see experiments with the original SlotAttention module, to see whether this improvement only applies to the SLATE model or is generally applicable for iterative inference in object-centric methods.

### Final thoughts
Overall I think this is a valid workshop paper, as it shows that a simple (in terms of lines of code) change can improve training of the Slot Attention module. Therefore I recommend acceptance.

---

### Official Review · Reviewer_VviG · 2022-03-14
**Review: Object Representations as Fixed Points**

**Rating:** 3
**Confidence:** 2

**Review:**

**Summary**: The authors present a method for training slot-attention based models using implicit differentiation by leveraging the interpretation of the slot-attention mechanism as a fixed-point operation. They demonstrate that the training algorithm is effective at stabilizing training, thereby improving state of the art model performance on multiple standard benchmarks while simultaneously making training of such models easier.

**Strong points**:
- The contribution of the paper is theoretically grounded and experiments tracking the jacobian of the transformation align well with theory.
- The method is simple and may have the potential to significantly improve the stability of slot-attention based models.

**Suggestions for minor improvements**:
- A greater number of seeds, specifically 2 seeds for visualization of vanilla slot attention and 3 seeds for the new method are not entirely convincing for the significant claims being made, especially when one of the three seeds does not agree. Furthermore the performance of the vanilla SLATE model on the presented two seeds is somewhat concerning given the extremely poor performance in comparison to prior published work.
- Remove some excessive claims without justification or further details. For example in the conclusion stating it may “potentially [improve] any iterative inference algorithm”.

Overall the paper appears to be a solid contribution to slot-attention based algorithms and I believe it would be a valuable addition to the workshop.

---

### Decision · Program_Chairs · 2022-03-24

**Decision:**

Accept (Oral)

**Comment:**

Both reviewers agree that this paper proposes a method that is simple and novel. It's theoretically grounded and fits the theme of the workshop. Thus, this paper is given an oral presentation at the workshop. Congratulations!

The authors are encouraged to take the points raised by reviewers into account when preparing the camera-ready version.